# Calibration of individual-based models to epidemiological data: A systematic review

C. Marijn Hazelbag[1]*, Jonathan Dushoff[1,2], Emanuel M. Dominic[1], Zinhle E. Mthombothi[1], Wim Delva[1,3,4,5,6,7]

1 South African DSI-NRF Centre of Excellence in Epidemiological Modelling and Analysis (SACEMA), Stellenbosch University, Stellenbosch, South Africa, 2 Department of Biology, Department of Mathematics and Statistics, Institute for Infectious Disease Research, McMaster University, Hamilton, Ontario, Canada, 3 School for Data Science and Computational Thinking, Stellenbosch University, Stellenbosch, South Africa, 4 Center for Statistics, I-BioStat, Hasselt University, Diepenbeek, Belgium, 5 Department of Global Health, Faculty of Medicine and Health, Stellenbosch University, Stellenbosch, South Africa, 6 International Centre for Reproductive Health, Ghent University, Ghent, Belgium, 7 Rega Institute for Medical Research, KU Leuven, Leuven, Belgium

* marijnhazelbag@sun.ac.za

**Data Availability Statement:** The data can be found on Dryad using https://datadryad.org/stash/dataset/doi:10.5061/dryad.8sf7m0cj6. The doi for the data is: doi:10.5061/dryad.8sf7m0cj6.

## Abstract

Individual-based models (IBMs) informing public health policy should be calibrated to data and provide estimates of uncertainty. Two main components of model-calibration methods are the parameter-search strategy and the goodness-of-fit (GOF) measure; many options exist for each of these. This review provides an overview of calibration methods used in IBMs modelling infectious disease spread. We identified articles on PubMed employing simulation-based methods to calibrate IBMs informing public health policy in HIV, tuberculosis, and malaria epidemiology published between 1 January 2013 and 31 December 2018. Articles were included if models stored individual-specific information, and calibration involved comparing model output to population-level targets. We extracted information on parameter-search strategies, GOF measures, and model validation. The PubMed search identified 653 candidate articles, of which 84 met the review criteria. Of the included articles, 40 (48%) combined a quantitative GOF measure with an algorithmic parameter-search strategy– either an optimisation algorithm (14/40) or a sampling algorithm (26/40). These 40 articles varied widely in their choices of parameter-search strategies and GOF measures. For the remaining 44 (52%) articles, the parameter-search strategy could either not be identified (32/44) or was described as an informal, non-reproducible method (12/44). Of these 44 articles, the majority (25/44) were unclear about the GOF measure used; of the rest, only five quantitatively evaluated GOF. Only a minority of the included articles, 14 (17%) provided a rationale for their choice of model-calibration method. Model validation was reported in 31 (37%) articles. Reporting on calibration methods is far from optimal in epidemiological modelling studies of HIV, malaria and TB transmission dynamics. The adoption of better documented, algorithmic calibration methods could improve both reproducibility and the quality of inference in model-based epidemiology. There is a need for research comparing the performance of calibration methods to inform decisions about the parameter-search strategies and GOF measures.

**Funding:** WD was supported by grant 12L5816N from the Research Foundation – Flanders (FWO). The funder had no role in study design, data collection and analysis, decision to publish, or preparation of the manuscript.

**Competing interests:** The authors have declared that no competing interests exist.

## Author summary

Calibration—that is, "fitting" the model to data—is a crucial part of using mathematical models to better forecast and control the population-level spread of infectious diseases. Evidence that the mathematical model is well-calibrated improves confidence that the model provides a realistic picture of the consequences of health policy decisions. To make informed decisions, Policymakers need information about uncertainty: i.e., what is the range of likely outcomes (rather than just a single prediction). Thus, modellers should also strive to provide accurate measurements of uncertainty, both for their model parameters and for their predictions. This systematic review provides an overview of the methods used to calibrate individual-based models (IBMs) of the spread of HIV, malaria, and tuberculosis. We found that less than half of the reviewed articles used reproducible, non-subjective calibration methods. For the remaining articles, the method could either not be identified or was described as an informal, non-reproducible method. Only one-third of the articles obtained estimates of parameter uncertainty. We conclude that the adoption of better-documented, algorithmic calibration methods could improve both reproducibility and the quality of inference in model-based epidemiology.

## Introduction

Individual-based models (IBMs) intended to inform public health policy should be calibrated to real-world data and provide valid estimates of uncertainty [1], [2]. IBMs track information for a simulated collection of interacting individuals [3]. IBMs allow for more detailed incorporation of heterogeneity, spatial structure, and individual-level adaptation (e.g. physiological or behavioural changes) compared to other modelling frameworks [4]. This complexity makes IBMs valuable planning tools, particularly in settings where real-world intricacies that are not accounted for in simpler models have important effects [5], [6]. However, researchers and policymakers often battle with the question of how much value they can attach to the results of IBMs [7]. Fitting an IBM to empirical data (calibration) improves confidence that the simulation model provides a realistic and accurate estimate of the outcome of health policy decisions (e.g. projection of the disease prevalence under different intervention strategies, or the cost-effectiveness of different intervention strategies) [8]–[12]. Transparent reporting on calibration methods for IBMs is therefore required [11], [12].

Parameter values with accompanying confidence intervals used in IBMs are obtained from the literature and are often obtained through statistical estimation. When researchers cannot estimate parameters from empirical data, they obtain their likely values through calibration [12]. Parameter calibration is often difficult for IBMs because their greater complexity can render the likelihood function analytically intractable (i.e. it is impossible to write down the likelihood function in closed form) or prevent explicit numerical calculation of the likelihood function [13]–[15]. Consequently, simulation-based calibration methods that avoid the use of a likelihood function in closed form have been developed [16]. These methods run the model for different parameter sets to identify parameter sets producing model output that best resembles the summary statistics obtained from the empirical data (e.g. disease prevalence over time). Formal simulation-based calibration requires *summary statistics* (*targets*) from empirical data, a *parameter-search strategy* for exploring the parameter space, a *goodness-of-fit (GOF)* measure to evaluate the concordance between model output and targets, *acceptance criteria* to determine which parameter sets produce model output close enough to the targets, and a *stopping rule* to determine when

the calibration ends [9][17]. IBMs vary in their complexity (i.e. the number of parameters) and the amount of data available for calibration and validation [10]. Simulation-based calibration of IBMs of higher complexity is typically more computationally intensive [18], [19].

In this review, we pay particular attention to the parameter-search strategy and GOF measure used. Algorithmic parameter-search strategies can be divided into *optimisation algorithms* and *sampling algorithms* [14], S2 Table describes commonly used algorithms. Optimisation algorithms find the parameter combination that optimises the GOF, resulting in a single best parameter combination. Examples include grid-search and iterative, descent-guided optimisation algorithms using simplex-based or direct search methods (e.g. the Nelder-Mead method) [20], but many different algorithms exist [21]. Optimisation algorithms provide only point estimates of parameters; once these are found, another algorithm may be used to obtain confidence intervals (e.g. the profile likelihood method, Fisher information, etc.) [22], [23]. Sampling algorithms aim to find a distribution of parameter values that approximate the likelihood surface or posterior distribution. Examples include approximate Bayesian computation (ABC) methods and sampling importance resampling [8], [13], [14], [24], [25]. Parameter distributions obtained from sampling algorithms allow for the representation of correlations between parameters and for parameter uncertainty to be incorporated into model projections [2], [6], [8], [17], [26]. Quantitative measures of GOF include distance measures (e.g. relative distance, squared distance) and measures based on a surrogate likelihood function: the likelihood of observing the target statistic under the assumption that the model output is a random draw from a presumed distribution (e.g. binomial for prevalence statistics). As the model output is not necessarily distributed as presumed, we refer to this likelihood as the "surrogate" likelihood. A more subjective method of calibration involves the manual adjustment of parameter values, followed by a visual assessment of whether the model outputs resemble empirical data [27].

Previous research in the context of IBMs of HIV transmission found that 22 (69%) out of 32 included articles described the process through which the model was calibrated to data [12]. The impact of stochasticity on the model results, defined as the random variation in model output induced by running the model multiple times using the same parameter value with a different random seed, was summarised in nearly half (15/32) of the articles [12]. The depth of reporting on calibration methods was highly variable [9], [12]. A systematic review in the context of population-level health policy models, including 37 articles, found that 25(71%) of these performed model calibration [28]. About half (12/25) of these articles reported on the calibration methods used, whereas the other half (13/25) used informal methods for parameter calibration or did not report on the calibration methods [28]. Previous research on calibration methods in cancer-simulation models in general–not IBMs specifically–found that 131 (85%) out of 154 included articles may have calibrated at least one unknown parameter. Of the 131 articles that calibrated parameters, the majority (84/131) did not describe the use of a GOF measure, the rest either used a quantitative GOF (27/131) such as the likelihood or distance measures or used visual assessment of GOF (20/131) [9]. Only a few articles reported parameter distributions resulting from calibration; most only presented a single best parameter combination [9]. Information on the parameter-search strategy and stopping rules was generally not well described, and acceptance criteria were rarely mentioned [9], [29]. Of the 154 articles included in the review by Stout *et al.*, 80 (52%) mentioned model validation [9]. However, while previous studies have reviewed specific portions of the modelling literature, they either did not focus on IBMs or did not focus on the calibration methods in much detail.

We conducted a systematic review of epidemiological studies using IBMs of the HIV, malaria and tuberculosis (TB) epidemics, as these have been among the most investigated epidemics with the highest global burden of disease [30]. We aim to provide an overview of current practices in the simulation-based calibration of IBMs.

## Results

### Selection of articles for inclusion

The PubMed search resulted in 653 publications, of which 84 articles were included for review; 388 were excluded based on title and abstract, and another 181 were excluded based on a full-text review (see Fig 1). The number of articles selected by publication year increased from seven in 2013 to 20 in 2018.

### Scope and objectives of included articles

S1 Table summarises the characteristics of the included articles. Fifty-eight (69%) of the included articles presented IBMs in HIV research, 16 (19%) concerned malaria, and another 10 (12%) concerned tuberculosis.

Most articles, namely 56 (67%), investigated the effect of an intervention, 17 articles looked at behavioural or biological explanations for the observed epidemic, and other goals (e.g. parameter estimation, model development) were used in 17. In total, six (7%) articles had two objectives. For most of these (5/6), one of the objectives was investigating the effect of an intervention (see S1 Table).

### Parameter-search strategies and measures of GOF

Of the included articles, 40 (48%) combined a quantitative measure of GOF with an algorithmic parameter-search strategy, which was an optimisation algorithm (14/40) or a sampling

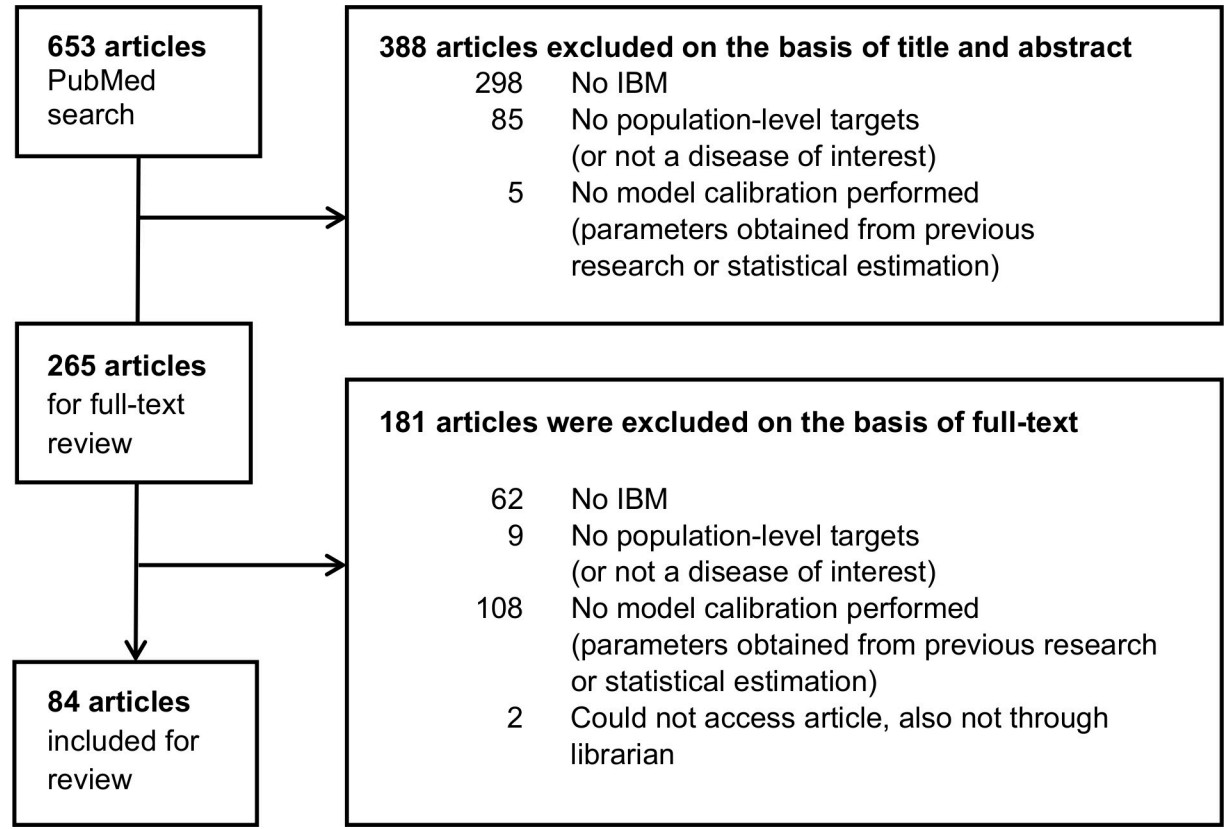

**Fig 1. PRISMA flow diagram detailing the selection process of articles included in the review.**

algorithm (26/40) (see Fig 2). For the remaining 44 (52%) articles, the parameter-search strategy could either not be identified (32/44) or was described as an informal, non-reproducible method (12/44). Tables A, B and C in S1 Appendix show that there is no convincing evidence that the parameter search strategy changed with publication year or differed by disease studied. A brief description of the methods referred to in Fig 2 under optimisation algorithm and sampling algorithm is provided in S2 Table.

Detailed information on calibration methods for the 14 (17%) articles using optimisation algorithms is reported in Table 1. For the parameter-search strategy, most articles used either a grid search (7/14), Latin square (1/14) or random draw from tolerable range (1/14), followed by the selection of the single best parameter combination. Several iterative, descent-guided optimisation algorithms (i.e. Nelder-Mead, interior-point algorithm, coordinate descent with golden section search, random search mechanism) were used in the remaining articles (5/14). Of these five articles, most (4/5) accepted a single best parameter combination without confidence intervals, while the remaining article obtained confidence intervals around parameter estimates (see S1 Text.). For the GOF measure, the most common choice was a squared distance (6/14). Various GOF measures were used in the remaining articles; these include absolute distances (2/14) and R-squared (2/14).

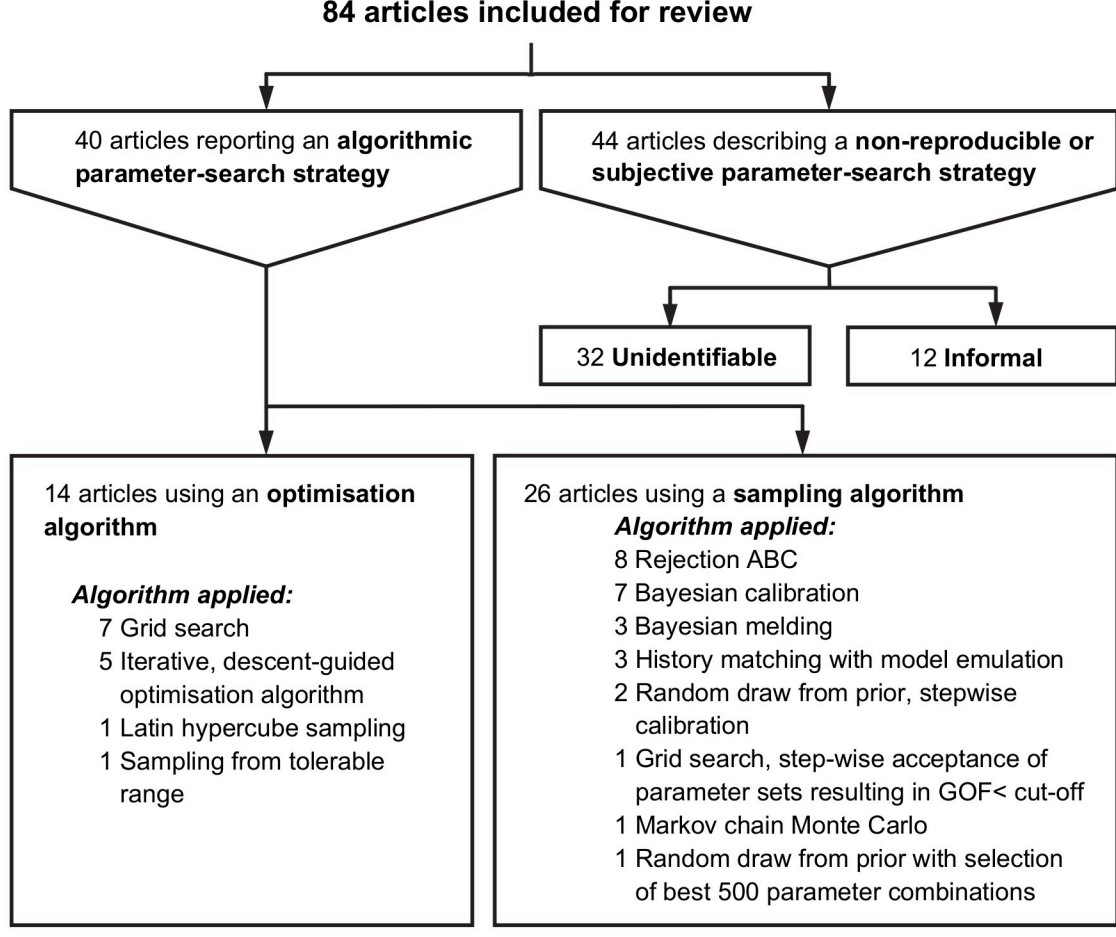

**Fig 2. Reporting and application of parameter search strategies in epidemiological studies.**

**Table 1. Details of the calibration methods used in articles using optimisation algorithms for calibration, sorted by parameter search strategy algorithm.**

| Authors | Year | Pathogen | Parameter search strategy algorithm | GOF |
|---|---|---|---|---|
| **Luo** *et al.* | 2018 | HIV | Grid search | Absolute distance |
| **Romero-Severson** *et al.* | 2013 | HIV | Grid search | Kolmogorov-Smirnov |
| **Marshall** *et al.* | 2018 | HIV | Grid search | R-squared |
| **Goedel** *et al.* | 2018 | HIV | Grid search | R-squared and Manhattan distance of parameters |
| **Brookmeyer** *et al.* | 2014 | HIV | Grid search | Squared distance |
| **Suen** *et al.* | 2014 | TB | Grid search | Number of model outputs within the confidence intervals around the targets |
| **Suen** *et al.* | 2015 | TB | Grid search | Number of model outputs within the confidence intervals around the targets |
| **Bershteyn** *et al.* | 2013 | HIV | Iterative, descent-guided optimisation algorithm (*Coordinate descent w. golden section search*) | Squared distance |
| **Klein** *et al.* | 2015 | HIV | Iterative, descent-guided optimisation algorithm (*Coordinate descent w. golden section search*) | Squared distance |
| **Sauboin** *et al.* | 2015 | Malaria | Iterative, descent-guided optimisation algorithm (*Interior point algorithm*, *hill-climbing*) | Squared distance |
| **Knight** *et al.* | 2015 | TB, HIV | Iterative, descent-guided optimisation algorithm (*Nelder-Mead*) | Squared distance |
| **Kasaie** *et al.* | 2018 | HIV | Iterative, descent-guided optimisation algorithm (*Random search mechanism*) | Absolute distance |
| **Shrestha** *et al.* | 2017 | TB | Latin hypercube sampling | Surrogate likelihood |
| **Jewell** *et al.* | 2015 | HIV | Sampling from tolerable range | Squared distance |

Table 2 contains the details of the calibration methods in the 26 (31%) articles using sampling algorithms. Random sampling from the prior, followed by rejection ABC, was used the most (8/26). Different types of Bayesian calibration (7/26), Bayesian melding (3/26) and history matching with model emulation (3/26) were also used. Most articles (10/26) used the surrogate likelihood as a measure of GOF, and Various GOF measures were used in the remaining articles, these include absolute distances (4/26), relative distances (4/26) and squared distances (4/26). (see Table 2).

From the 44 (52%) articles with unidentifiable or informal parameter-search strategies, the majority (25/44) are also unclear about the GOF used, while the rest either relied on visual inspection as a GOF (14/44) or used a quantitative GOF (5/44).

Only 14 (17%) of the 84 included articles provided a rationale for their choice of model-calibration method. For example, McCreesh *et al.* [31] reported: "The model was fitted to the empirical data using history matching with model emulation, which allowed uncertainties in model inputs and outputs to be fully represented, and allowed realistic estimates of uncertainty in model results to be obtained" (see S2 Text. for more examples). Other examples indicate that an algorithmic calibration method failed to provide either a good fit or parameter estimates: "Ultimately, we chose to use visual inspection because the survival curves did not fit closely enough using the other two more quantitative approaches." [32] Or "[Calibration] was unable to resolve co-varying parameters. These parameters were adjusted by hand. . ." [33].

Ten out of the 84 articles included (12%) used a weighted calculation of GOF. Four articles weighted the GOF based on the amount of data behind the summary statistic fitted to, for example by weighting based on the inverse of the width of the confidence interval around the data. In contrast, one article increased the weight for a data source for which fewer data was available. Other strategies included weighting based on a subjective assessment of the quality of the data, or weighting based on which data they wanted the model to fit best. One article

**Table 2. Details of the calibration methods in articles using sampling algorithms for calibration, sorted by parameter search strategy algorithm.**

| Authors | Year | Pathogen | Parameter search strategy algorithm | GOF |
|---|---|---|---|---|
| **Cameron** *et al.* | 2015 | Malaria | Bayesian calibration (*Combining model emulation with MCMC*) | Surrogate likelihood |
| **Huynh** *et al.* | 2015 | TB | Bayesian calibration (*Latin hypercube with IMIS*) | Surrogate likelihood |
| **Chang** *et al.* | 2018 | TB | Bayesian calibration (*Latin hypercube with IMIS*) | Surrogate likelihood |
| **Penny** *et al.* | 2015 | Malaria | Bayesian calibration (*MCMC*) | Surrogate likelihood |
| **Penny** *et al.* | 2015 | Malaria | Bayesian calibration (*MCMC*) | Surrogate likelihood |
| **White** *et al.* | 2018 | Malaria | Bayesian calibration (*MCMC*) | Surrogate likelihood |
| **Schalkwyk** *et al.* | 2018 | HIV | Bayesian calibration (*Random draw from prior with SIR*) | Surrogate likelihood |
| **Abuelezam** *et al.* | 2016 | HIV | Bayesian melding | Squared distance |
| **McCormick** *et al.* | 2014 | HIV | Bayesian melding | Surrogate likelihood |
| **McCormick** *et al.* | 2017 | HIV | Bayesian melding | Surrogate likelihood |
| **Ciaranello** *et al.* | 2013 | HIV | Grid search, step-wise acceptance of parameter sets resulting in GOF < cut-off | Absolute distance |
| **McCreesh** *et al.* | 2017 | HIV | History matching with model emulation | Implausibility measure |
| **McCreesh** *et al.* | 2017 | HIV | History matching with model emulation | Implausibility measure |
| **McCreesh** *et al.* | 2018 | HIV | History matching with model emulation | Implausibility measure |
| **Shcherbacheva** *et al.* | 2018 | Malaria | Markov chain Monte Carlo | Absolute distance |
| **Johnson** *et al.* | 2016 | HIV | Random draw from prior with selection of best 500 parameter combinations | Surrogate likelihood |
| **Pizzitutti** *et al.* | 2015 | Malaria | Random draw from prior, stepwise calibration | Absolute distance |
| **Pizzitutti** *et al.* | 2018 | Malaria | Random draw from prior, stepwise calibration | Squared distance |
| **Nakagawa** *et al.* | 2016 | HIV | Rejection ABC (*Random draw from prior*) | Relative distance |
| **Nakagawa** *et al.* | 2017 | HIV | Rejection ABC (*Random draw from prior*) | Chi-square |
| **Cambiano** *et al.* | 2018 | HIV | Rejection ABC (*Random draw from prior*) | Relative distance |
| **Hontelez** *et al.* | 2013 | HIV | Rejection ABC (*Random draw from prior*) | Squared distance |
| **Phillips** *et al.* | 2013 | HIV | Rejection ABC (*Random draw from prior*) | Relative distance |
| **Phillips** *et al.* | 2015 | HIV | Rejection ABC (*Random draw from prior*) | Relative distance |
| **Shrestha** *et al.* | 2017 | HIV | Rejection ABC (*Random draw from prior*) | Absolute distance |
| **Tuite** *et al.* | 2017 | TB | Rejection ABC (*Random draw from prior*) | Squared distance |

IMIS, Incremental-mixture importance sampling; SIR, Sampling importance resampling; MCMC, Markov chain Monte Carlo.

down-weighted particular data to improve fit. Others stressed the importance of determining weights a priori since weights are chosen subjectively.

## Acceptance criteria and stopping rules

None (0/14) of the articles applying optimisation algorithms mentioned the acceptance criteria or stopping rules. Acceptance criteria and stopping rules applied in studies using sampling algorithms can be summarised as running the model until obtaining an arbitrary number of accepted parameter combinations.

## The number of target statistics, the number of calibrated parameters and the size of the simulated population

The number of target statistics was explicitly mentioned in only three (3%) of the 84 included articles, for 62 (74%) articles we had enough information to attempt to deduce this number from either text or figures. The remaining 19 (23%) articles either provided incomplete information (11/19) or no information (8/19). Some (4/65) of the articles for which we were able to obtain the number of target statistics had different numbers of target statistics for calibration in different locations or calibration to different diseases. The 61 (73%) articles for which we

were able to obtain a single count had a median number of target statistics of 23 (range 1–321). A histogram of the number of target statistics is provided in figure A in S2 Appendix. The number of target statistics differed between parameter search strategies (See Fig 3B, Kruskal-Wallis chi-square = 8.610, p = 0.035), with articles using sampling strategies having more target statistics compared to articles for which we could not identify the parameter search strategy (Wilcoxon rank-sum, Benjamini-Hochberg adjusted p-value = 0.025).

The number of calibrated parameters was explicitly mentioned in 11 (13%) of the 84 included articles, for another 53 (63%) articles it was possible to deduce this number from either text or figures. The remaining 20 (24%) articles either provided incomplete information (10/20) or no information at all (10/20). The 64 (75%) articles for which we were able to obtain a count had a median number of calibrated parameters of 10 (range 1–96). A histogram of the number of calibrated parameters is provided in figure B in S2 Appendix. The number of calibrated parameters differed between parameters search strategies (See Fig 3A, Kruskal-Wallis chi-square = 9.304, p = 0.026), with articles using sampling strategies having higher numbers of calibrated parameters compared to articles for which we could not identify the parameter search strategy (Wilcoxon rank-sum, Benjamini-Hochberg adjusted p-value = 0.050).

For 55 (66%) articles, we obtained counts for both the number of target statistics and the number of calibrated parameters. For many of these articles (17/55), the number of calibrated parameters appeared to exceed the number of target statistics. A plot of the number of target statistics against the number of calibrated parameters is provided in figure C in S2 Appendix.

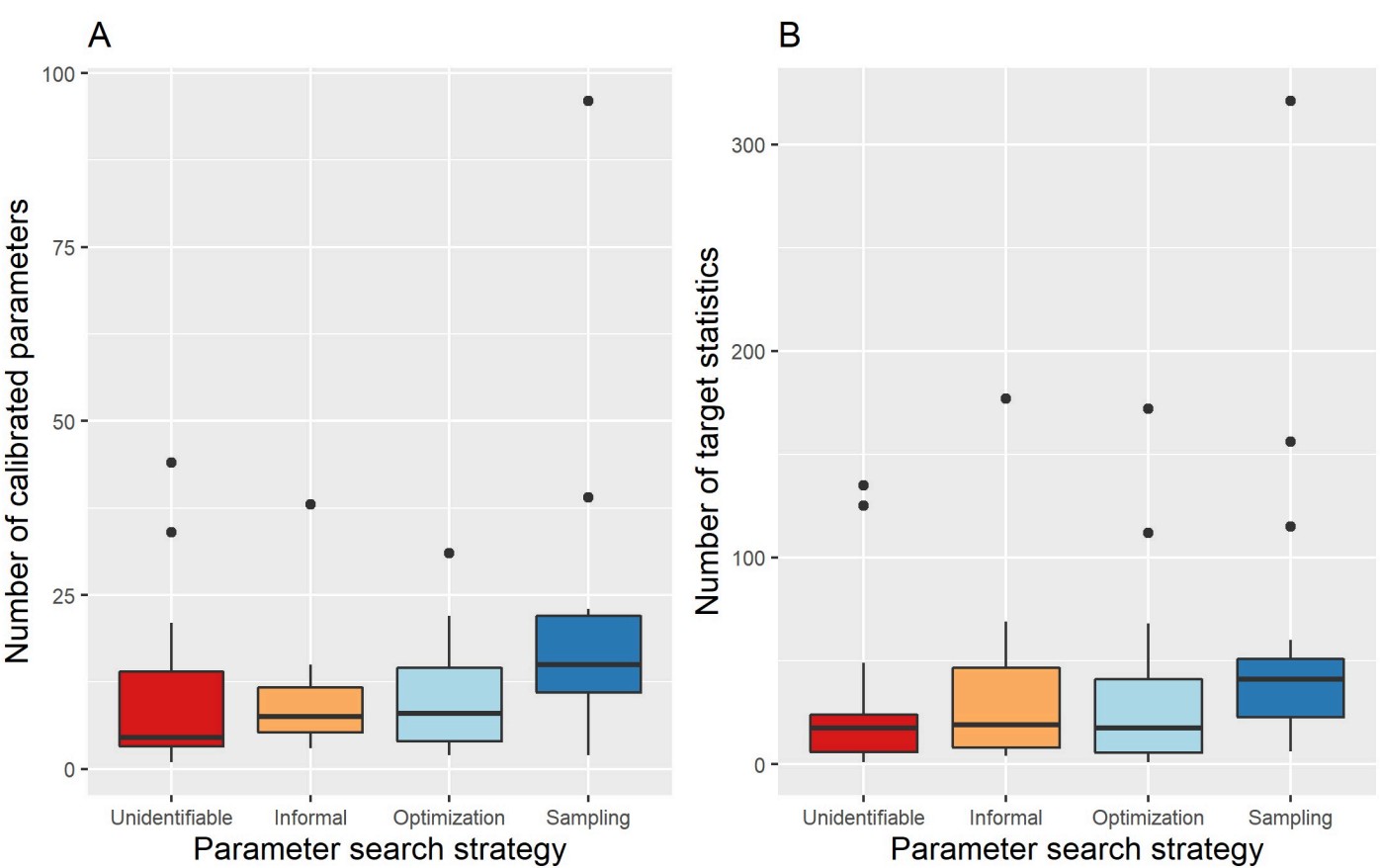

**Fig 3. Comparison of the number of calibrated parameters and target statistics between different parameter search strategies.** (A) Boxplots of the number of calibrated parameters for different parameter search strategies. (B) Boxplots of the number of target statistics for different parameter search strategies.

The size of the simulated population was explicitly mentioned in 54 (64%) of the 84 included articles, for another 9 (11%) articles it was possible to deduce this number from either text or figures. The remaining 21 (25%) articles either provided incomplete information (3/21) or no information at all (18/21). For the 63 (75%) articles for which we obtained a number, the median population size was 78000 (range: 250–47000000). A histogram of the $\log_{10}$ of the size of the simulated population is provided in figure D in S2 Appendix.

## Computational aspects and the use of platforms

The software used to build IBM was not reported in 33 (39%) of the articles. Sixteen articles (19%) used the low-level programming language C++, six (7%) used MATLAB, and another six (7%) used Python. Various other computing platforms were used in the remaining 23 (28%) articles. A high-performance computing facility was used in 16 (19%) articles.

Several simulation tools (i.e. CEPAC [34], EMOD [35] HIV-CDM [36], MicroCOSM [37], PATH [38], STDSIM [39] and TITAN [40]) were used in the articles modelling HIV. Similarly, two platforms (i.e. EMOD [41] and OpenMalaria [42]) were used in the articles modelling malaria. In the articles modelling tuberculosis, the only tool reported was EMOD [43].

## Model validation

Only 31 (37%) articles mentioned that a validation of the model had been performed.

## Discussion

More than half of IBMs we studied used non-reproducible or subjective calibration methods. Articles that reported the use of formal calibration methods used a wide range of parameter-search strategies and GOF measures. Only one-third of articles used calibration methods that quantify parameter uncertainty. These findings are important because choices concerning the calibration method can have substantial effects on model results and policy implications [2], [6]–[8], [44]–[46].

We encourage authors to use the standardised Calibration Reporting Checklist of Stout *et al.* [9]. Additionally, we propose an extended checklist in S3 Appendix based on the work presented in this paper. While algorithmic parameter-search strategies are in principle reproducible, unclear or incomplete reporting, and non-disclosure of software code can render them de facto non-reproducible. [47]. Manual adjustment of parameter values and visual inspection of GOF may perform equally well compared to other methods in terms of GOF alone [48], may provide researchers with valuable insights into and familiarity with the model [49], and can be useful for purely didactic purposes [50]–[52]. However, we advise against using these methods in analyses intended to inform public health as they do not favour reproducibility and involve subjective judgment, which may produce less than optimal calibration results and usually leads to the acceptance of a single parameter set (i.e. does not provide parameter uncertainty) [17]. On occasion, authors justified their choice of an informal method by indicating that algorithmic calibration methods did not converge to provide parameter estimates or failed to provide a satisfactory fit to the targets. A potential explanation for non-convergence of an algorithmic calibration method is that the parameters in question are unidentifiable, which is the case when a vast array of different parameter combinations provide a comparably good fit to the target statistics. Performing manual calibration in such an instance will deliver one set of parameters out of all of the parameter combinations that provide a fit. However, using this single parameter combination hides the fact that there is not enough information to uniquely identify the best parameter values. Furthermore, model-stochasticity provides the possibility that a great fit is found by chance for a parameter

combination for which the probability of observing the target statistics is lower than for other parameter combinations.

There are several methodological challenges in the calibration of individual-based models, including the choice of calibration method–i.e. the combination of algorithmic parameter-search strategy and GOF measure. The findings of the current review and previous research suggest that there is no consensus on which calibration method to use [9], [10], [17], [53], [54]. Additionally, some of the articles reviewed here indicated that algorithmic calibration methods had failed, leading the researchers to calibrate the model, either fully or partially, by hand. These issues suggest that there is a need for research comparing the performance of calibration methods to inform the choice of parameter-search strategy and GOF [10]. Previous research on calibration methods focused on the GOF [27], computation time and analyst time [48]. Where applicable, correct estimation of the posterior [55] should be a core aspect of performance. We further suggest investigating several contextual variables, including the amount and nature of the empirical data to calibrate against, the number and type of model parameters to be calibrated and insights to be derived from the calibrated model. As evident from our review, these contextual variables vary widely across IBM studies in epidemiology.

Another methodological challenge in the calibration of IBMs is determining a priori whether the target statistics provide sufficient information to calibrate the parameters [56], especially when the model has many parameters [57]. Firstly, the target statistics are based on variable amounts of raw data. Secondly, a time series of target statistics is often used, typically violating the assumption of independence implied by many calibration methods. Thirdly, the complexity of the model may hamper an appropriate specification of a prior parameter-distribution (including the specification of a correlation between parameters) that is fully informed by prior knowledge of the data-generating processes represented by the model. These problems preclude the use of standard statistical methods for calculating the number of target statistics that is sufficient for parameter calibration. A related problem is that target summary statistics are based on data from different sources, including observational data that are potentially affected by treatment-confounder feedback (e.g. time-dependent confounder CD4 cell count affected by prior cART treatment) [58]. Another related problem is that of validation, i.e. testing model performance on data that was not included in the calibration step. There is considerable debate on when data should be reserved for this purpose [54].

The last methodological aspect of IBMs we would like to draw attention to is the size of the simulated population [1], [59]. Intuitively, one would recommend that the simulated population size should be similar to the size of the population from which the samples were drawn that gave rise to the target statistics. However, for many studies, modelling the full population is not feasible with currently available computational infrastructure. Instead, researchers often adjust for the inflated stochasticity in the modelled system by averaging outcomes of interest over multiple simulation runs per parameter set [59]. How choices around modelled population size and analysis of model output affect the validity of model inference deserves further attention in future research.

Our results in the setting of HIV, TB and malaria IBMs indicate that the use of formal calibration methods (48% of articles) is higher than in previous research on simulation models in general–not IBMs specifically. Previously, only one-fifth to one-third of articles reporting on epidemiological models used a quantitative GOF [9], [60]. Our results concerning parameter uncertainty are also optimistic compared to previous research by Stout *et al.* on calibration methods in cancer models, which found that almost no articles quantified parameter uncertainty, but instead accepted a single best-fitting parameter set as the result of the calibration [9]. The same researchers reported that several different combinations of parameter-search strategies and GOFs were used [9], outcomes which are similar to our findings. Stout *et al.*

report that articles rarely describe acceptance criteria and stopping rules. Stout *et al.* also report that a standard description of the calibration process lacks in almost all articles [9]. Similarly, previous research on IBMs of HIV transmission found that reporting was lacking in the description of calibration methods [12]. All of this is in agreement with the results of the current review. Concerning the goals of the included articles, our results broadly agree with Punyacharoensin *et al.* They found that the main goals of HIV transmission models for the study of men who have sex with men are: making projections for the epidemic, investigating how the incorporation of various assumptions around the behavioural or biological characteristics affect these projections, and evaluating the impact of interventions [60].

To our knowledge, this is the first detailed review of methods used to calibrate IBMs of HIV, malaria and TB epidemics. A limitation of our study is that we are unsure to what extent the results are generalisable to other infectious diseases. We encourage future research on other diseases to confirm or refute our current findings on the use of and reporting on methods in the calibration of IBMs in epidemiological research. Similarly, since our PubMed search excluded articles matching "molecular", we may have missed relevant articles. However, we don't believe this selection is likely to bias the findings of this review. Another possible concern is that we don't control for overlaps in authorship; thus, we effectively treat articles that come from a given"research group" as independent observations, even though the calibration method used by a particular group is often the same, as we show in Tables 1 and 2. Another limitation is that the counts presented in this review often had to be deduced from the article, this was a difficult and laborious task involving manual counting of target statistics in either the text, figures or tables, a process that is prone to error. A final limitation is that we did not go into the strengths and weaknesses of each method. Existing literature compares the performance of alternative algorithms for calibrating the same model but does not allow us to draw general conclusions [10]. As a starting point for comparison, we provide a brief description of calibration methods in S2 Table.

In conclusion, it appears that calibrating individual-based models in epidemiological studies of HIV, malaria and TB transmission dynamics remains more of an art than a science. Besides limited reproducibility for a majority of the modelling studies in our review, our findings raise concerns over the correctness of model inference (e.g., estimated impact of past or future interventions) for models that are poorly calibrated. The quality of inference and reproducibility in model-based epidemiology could benefit from the adoption of algorithmic parameter-search strategies and better-documented calibration and validation methods. We recommend the use of sampling algorithms to obtain valid estimates of parameter uncertainty and correlations between parameters. There is a need for simulation-based studies that compare the performance, strengths and limitations of different methods for calibrating IBMs to epidemiological data.

## Materials and methods

This review was performed following the Preferred Reporting Items for Systematic Reviews and Meta-Analyses (PRISMA) statement [61]. The PRISMA flow diagram details the selection process of articles included for review (see Fig 1).

### Search strategy and selection criteria

We identified articles on PubMed that employed simulation-based methods to calibrate IBMs of HIV, malaria and tuberculosis, and that were published between 1 January 2013 and 31 December 2018. Six years seemed to be long enough to yield a sizeable amount of information and to observe recent time trends, and short enough to be feasible and to speak to recent

practices in model calibration in epidemiological modelling studies. The following search query was performed on 31 January 2019: '((HIV[tiab] OR malaria[tiab] OR tuberculo*[tiab] OR TB[tiab]) AND (infect* OR transmi* OR prevent*) AND (computer simulation[tiab] OR microsimulation[tiab] OR simulation[tiab] OR agent-based[tiab] OR individual-based[tiab] OR computer model*[tiab] OR computerized model*[tiab]) AND ("2013/01/01"[Date—publication]: "2018/12/31"[Date—publication]) NOT(molecular))'.

Eligibility criteria were agreed upon by WD, JD and CMH before screening. Articles were included if models stored individual-specific information and calibration involved running the model and comparing model output to population-level targets expressed as summary statistics. We excluded review articles, statistical simulation studies, and studies that focused on molecular biology and immunology because we were primarily interested in studies informing public health policy.

Titles and abstracts were screened for eligibility by CMH, and difficult cases were discussed with WD. If the title and abstract did not provide sufficient information for exclusion, a full-text examination was performed. Full-text inclusion was performed by two independent researchers (CMH and either ZM or ED) for a subset of 100 articles. CMH included 28 articles, of which ZM and ED did not include six; these six articles were double-checked by WD and consequently included for review. ZM included four articles that CMH did not include these four articles were double-checked by WD and consequently not included for review. After that, full-text inclusion was performed by CMH in consultation with WD.

## Data extraction

For each article, we extracted information on the objective of the study (i.e. estimating the effect of an intervention, investigating a behavioural or biological explanation for the observed infectious disease outbreak or other goals including estimation of parameters or model development), the parameter-search strategy and the GOF measure, the rationale for choosing this calibration strategy over alternatives, and model validation. Acceptance criteria and stopping rules are only relevant for articles applying algorithmic parameter-search strategies and collected for that subset of articles. For readability purposes, we say "used" to mean "reported the use of" throughout this review.

Information was collected independently by two reviewers (CMH and either ZM or ED) for each article included using a prospectively developed form. This form was based on the Calibration Reporting Checklist of Stout *et al.* [9] and was extended by several items, including; the software and hardware used to build the model, the size of the initial population of agents and the name of the modelling platform. Additionally, we inserted several items to collect information on the number of calibrated parameters, the number of fixed parameters, and the number of targets. We noted how information on these counts was reported in the articles (i.e. the number was explicitly provided, could be deduced from text or figures, was provided incompletely or was not provided).

Information on calibration methods was extracted verbatim, allowing for later classification. Articles on which there was disagreement in the classification were discussed by WD, JD and CMH until an agreement was reached. We classified articles reporting both algorithmic and informal calibration as informal since doing part of the calibration informally makes the entire calibration irreproducible.

## Statistical analysis

R 3.5.0 (www.r-project.org) was used to perform the statistical analyses [62]. Differences between groups in non-normally distributed continuous variables were analysed by the

nonparametric Kruskal-Wallis test [63]. Wilcoxon rank-sum test was used to determine which groups differed significantly [63]. Benjamini-Hochberg (BH) correction was used to adjust for multiple testing [64].

## Supporting information

**S1 Table. Articles included for review.**
(DOCX)

**S2 Table. Description of calibration algorithms.**
(DOCX)

**S1 Text. Obtaining parameter uncertainty using an optimisation algorithm, quoted from Sauboin et al.**
(DOCX)

**S2 Text. Selected quotes of rationales for choosing model calibration method.**
(DOCX)

**S1 Appendix. Parameter search strategies by disease and year of publication.**
(DOCX)

**S2 Appendix. Histograms and plots for counts of targets, calibrated parameters and the size of the simulated population.**
(DOCX)

**S3 Appendix. Calibration reporting checklist.**
(DOCX)

## Acknowledgments

The authors gratefully acknowledge the help of all SACEMA students and researchers, specifically the fruitful conversations and helpful comments on the manuscript by Prof. Alex Welte, Mrs Cari van Schalkwyk, Dr Florian Marx, Prof. Juliet Pulliam and Dr Larisse Bolton. We would also like to acknowledge Mrs Marisa Honey and Mrs Susan Lotz from the Stellenbosch writing lab, who copy-edited a first version of the manuscript.

## Author Contributions

**Conceptualization:** C. Marijn Hazelbag, Jonathan Dushoff, Wim Delva.

**Data curation:** C. Marijn Hazelbag, Emanuel M. Dominic, Zinhle E. Mthombothi, Wim Delva.

**Formal analysis:** C. Marijn Hazelbag.

**Investigation:** C. Marijn Hazelbag, Jonathan Dushoff, Emanuel M. Dominic, Zinhle E. Mthombothi, Wim Delva.

**Methodology:** C. Marijn Hazelbag, Jonathan Dushoff, Wim Delva.

**Project administration:** C. Marijn Hazelbag.

**Supervision:** Wim Delva.

**Visualization:** C. Marijn Hazelbag.

**Writing – original draft:** C. Marijn Hazelbag, Jonathan Dushoff, Emanuel M. Dominic, Zinhle E. Mthombothi, Wim Delva.

**Writing – review & editing:** C. Marijn Hazelbag, Jonathan Dushoff, Emanuel M. Dominic, Zinhle E. Mthombothi, Wim Delva.

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
