## [Decision Letter · Decision Letter 0]

6 Nov 2019

Dear Dr Hazelbag,

Thank you very much for submitting your manuscript 'Fitting individual-based models to data in HIV, tuberculosis and malaria: a systematic review' for review by PLOS Computational Biology. Your manuscript has been fully evaluated by the PLOS Computational Biology editorial team and in this case also by independent peer reviewers. The reviewers appreciated the attention to an important problem, but raised some substantial concerns about the manuscript as it currently stands. While your manuscript cannot be accepted in its present form, we are willing to consider a revised version in which the issues raised by the reviewers have been adequately addressed. We cannot, of course, promise publication at that time.

Sincerely,

Roger Dimitri Kouyos

Associate Editor

PLOS Computational Biology

Virginia Pitzer

Deputy Editor

PLOS Computational Biology

[LINK]

Reviewer's Responses to Questions

**Comments to the Authors:**

Reviewer #1: Dear authors,

Please find the attached review.

Reviewer #2: This systematic review used HIV, Malaria, and Tuberculosis studies to provide an overview of the fitting methods used in IBMs modelling infectious disease spread. This is critical as the usage of IBMs becomes more common nowadays than the deterministic models. The calibration of IBMs to data is often a challenge. Overall, this is an interesting and well-written paper. I have some comments that need to be clarified.

1. Line 98: In addition to the stochasticity, the number of parameters to estimate in IBM indicates the complexity of the parameter-search strategy (i.e. estimating only one parameter is the search on one-dimensional real space (1D), estimating two parameters is the search on two-dimensional real space (2D), and so on). The authors may comment in the introduction to show how the complexity of calibration varies between IBMs.

2. Line 144: It is better to provide a reference for the most investigated epidemics.

3. Line 155: Is there any reasons why the authors did the search from 2013?

4. Line 179: “the goal” need to be clarified along this line.

5. Fig 2 lists the different methods of parameter search strategy found, but without minimal explanations. It is better to include in the appendix a table that explains briefly these methods for non-modelers.

6. Line 258: “…while the rest either relied on visual inspection as a GOF (14 articles) or used a quantitative GOF (five articles)...” The authors should discuss why some studies use manual fitting instead of formal parameter-search strategies. This manual fitting is impossible in case of multi-fitting parameters. Another issue of manual fitting is not reproducible due to the stochasticity in IBM.

7. Line 300: A reference along this line is very useful for the simulation tools.

8. This systematic review provides interesting data about the methods of calibration of IBMs, but it did not provide a comparison, which method is best, or what is the strength and limitations of each method. The authors should clarify this point in the limitation section.

9. Minor comment: There is an issue in a cell in Table 1.

Reviewer #3: The authors did a tedious and very valuable work in order to identify articles that used an individual-based model (IBM) to fit to data in HIV, tuberculosis (TB) and malaria, and assess the proportion of them that reported the parameter-search strategy and the type of parameter-search strategy used. This work is particularly important as one aims for more transparency on the optimization methods used in modelling works. However, the authors could go a little bit further in order to answer questions such as:

• Does the proportion of articles reporting the parameter search strategy used vary according to the disease studied (HIV, TB, malaria)?

• What about uncertainty? Is the uncertainty related to the parameter-search strategy (e.g. confidence interval or credibility interval) reported? Does the proportion of article reporting uncertainty depend on the field (i.e. HIV, TB, malaria) or on the parameter-search strategy used (sampling or optimization strategy)?

• How many parameters are estimated in each study? Does this number depend on the parameter search strategy? Are the most complex models (i.e. the ones with the highest number of parameter to estimate) the ones that do not report uncertainty/search strategy?

Answering these questions could help better identify the articles that report searching method less frequently (e.g. assess whether it is related to the field studied). Reporting information about the number of parameters estimated and whether uncertainty is reported could also provide a wider understanding of the issue of lack of transparency that we face in some fields.

In addition, I have a few minor comments:

• Lines 49-54: This is mainly repetition of the results already presented in the previous paragraph. Consider removing this paragraph and replacing by a discussion of the results, e.g. something that looks like the second part of “Author summary” (lines 65-72).

• Line 98-99: To me, it is not clear why a greater complexity could make exact likelihood calculation impossible. I would rather say that greater complexity prevent from identifying the exact maximum likelihood estimator, but it should not prevent the model to calculate the exact likelihood.

• Line 129: What kind of stochasticity do you mention here? Would it not be relevant to report stochasticity in your systematic review as well?

• Line 155: Why only from 2013 and not before?

• Line 166: Maybe you could mention before (in abstract?) that you will focus on studies informing public health policy.

• Line 241: In the “GOF” column of Table 1, there is some formatting problem, as we can not read the whole text in the “Suen et al.” column.

• Line 271: It is not clear how you could obtain a weight (i.e. a number) with the ”inverse of the confidence interval” which are two numbers.

• Lines 309-323: Consider rewriting this paragraph, as, as it is now, it repeats what you already said in the “Results” section. I see the point of summarizing the results, but this could be in a more concise way.

• Line 377: Was it not 48% (40/84) before, instead of 49%?

**Have all data underlying the figures and results presented in the manuscript been provided?**

Reviewer #1: Yes

Reviewer #2: Yes

Reviewer #3: Yes

PLOS authors have the option to publish the peer review history of their article (what does this mean?). If published, this will include your full peer review and any attached files.

Reviewer #1: No

Reviewer #2: Yes: H.H. Ayoub

Reviewer #3: No

---

## [Decision Letter · Decision Letter 1]

10 Mar 2020

Dear Dr. Hazelbag,

Thank you very much for submitting your manuscript "Calibration of individual-based models to epidemiological data: a systematic review" for consideration at PLOS Computational Biology. As with all papers reviewed by the journal, your manuscript was reviewed by members of the editorial board and by several independent reviewers. The reviewers raised only one minor issue still requires attention. Based on the reviews, we are likely to accept this manuscript for publication, providing that you modify the manuscript according to the review recommendations.

Sincerely,

Roger Dimitri Kouyos

Associate Editor

PLOS Computational Biology

Virginia Pitzer

Deputy Editor

PLOS Computational Biology

[LINK]

Reviewer's Responses to Questions

**Comments to the Authors:**

Reviewer #1: Good job. I have no additional comments.

Reviewer #2: The authors have satisfactorily responded to all my questions and made the necessary changes to the manuscript.

Reviewer #3: All the previously reported issues have been addressed by the authors. The authors might however check the following issue. In Fig3, title (and legend) mentioned that the figure reports the number of target statistics, while the y-axis label mentions the number of calibrated parameters. This must be corrected. Additionally, the authors could present both figures (number of targets and number of calibrated parameters according to the parameter search strategy) side by side in the main manuscript.

**Have all data underlying the figures and results presented in the manuscript been provided?**

Reviewer #1: Yes

Reviewer #2: Yes

Reviewer #3: Yes

PLOS authors have the option to publish the peer review history of their article (what does this mean?). If published, this will include your full peer review and any attached files.

Reviewer #1: No

Reviewer #2: No

Reviewer #3: No
---

## [Editor Report · Decision Letter 2]

21 Apr 2020

Dear Dr. Hazelbag,

We are pleased to inform you that your manuscript 'Calibration of individual-based models to epidemiological data: a systematic review' has been provisionally accepted for publication in PLOS Computational Biology.

Best regards,

Roger Dimitri Kouyos

Associate Editor

PLOS Computational Biology

Virginia Pitzer

Deputy Editor

PLOS Computational Biology

---

## [Editor Report · Acceptance letter]

1 May 2020

PCOMPBIOL-D-19-01519R2 

Calibration of individual-based models to epidemiological data: a systematic review

Dear Dr Hazelbag,

I am pleased to inform you that your manuscript has been formally accepted for publication in PLOS Computational Biology. Your manuscript is now with our production department and you will be notified of the publication date in due course.

With kind regards,

Laura Mallard
